# Antiplatelet and Anticoagulant Activity of Isorhamnetin and Its Derivatives Isolated from Sea Buckthorn Berries, Measured in Whole Blood

**DOI:** 10.3390/molecules27144429

**Published:** 2022-07-11

**Authors:** Anna Stochmal, Agata Rolnik, Bartosz Skalski, Jerzy Zuchowski, Beata Olas

**Affiliations:** 1Department of Biochemistry, Institute of Soil Science and Plant Cultivation, State Research Institute, 24-100 Puławy, Poland; jzuchowski@iung.pulawy.pl; 2Department of General Biochemistry, Faculty of Biology and Environmental Protection, University of Lodz, 90-236 Lodz, Poland; agata.rolnik@edu.uni.lodz.pl (A.R.); bartosz.skalski@umed.lodz.pl (B.S.); olasb@biol.uni.lodz.pl (B.O.); 3Department of Medical Biotechnology, Medical University of Lodz, 90-752 Lodz, Poland

**Keywords:** sea buckthorn berries, isorhamnetin, antiplatelet potential, anticoagulant activity

## Abstract

Blood platelets play a crucial role in hemostasis, the process responsible for keeping blood flowing in the circulatory system. However, unnecessary platelet activation can lead to aggregation at the site of atherosclerotic plaque rapture and the formation of a thrombus, which promotes atherothrombotic diseases. Various dietary components, such as phenolic compounds, are known to demonstrate antiplatelet and anticoagulant properties, and it is possible that these could form an important element in the prophylaxis and therapy of cardiovascular diseases. Our present study examined the biological activity of isorhamnetin (**1**) and two isorhamnetin derivatives, (**2**): 3-*O*-beta-glucoside-7-*O*-alpha-rhamnoside and (**3**): 3-*O*-beta-glucoside-7-*O*-alpha-(3″′-isovaleryl)-rhamnoside, isolated from the phenolic fraction of sea buckthorn fruit, against human washed blood platelets and human whole blood in vitro. The anti-platelet and anticoagulant potential was determined using (A) flow cytometry, (B) the thrombus-formation analysis system (T-TAS) and (C) colorimetry. The results of the T-TAS test indicate that the AUC_10_ (Area Under the Curve) of the tested phenolic compounds (compounds **1**, **2** and **3**; 50 µg/mL) was markedly reduced compared to the control values. Moreover, flavonol demonstrated anti-platelet potential, including anti-adhesive activity, with these effects being more intense in compound **2** than isorhamnetin. Different actions of flavonol on platelet activation may depend on their binding ability to various receptors on blood platelets. However, the mechanism of their anti-platelet potential requires further additional studies, including in vitro and in vivo experiments.

## 1. Introduction

Blood platelets play a crucial role in hemostasis, a process responsible for keeping the blood flowing in the circulatory system. Platelets are generated from megakaryocytes in bone narrow, with daily production providing 150–400 × 10^9^ platelets per liter of blood. The cells display a range of surface receptors and adhesion molecules, and contain numerous granules; these are used to initiate coagulation cascades in primary hemostasis, where blood platelets adhere to the extracellular matrix [1]. Platelet activation may be stimulated by platelet secretion products and local prothrombotic factors, including tissue factors. There are two main pathways leading to platelet activation, but both result in the rapid binding of platelets to damaged blood vessels.

Stable platelet adhesion requires an interaction between glycoprotein GPVI, integrin-α_2_β_1,_ and collagen, and between integrin-α_5_β_1_ and fibronectin. The binding of collagen to GPVI results in platelet activation and the release of soluble agonists, including ADP and thromboxane A_2_, leading to the activation of GPIIb/IIIa, a major receptor for fibrinogen. As a result of this process, the platelets aggregate and form a thrombus [2]. However, unnecessary platelet activation can lead to the development of atherothrombotic diseases caused by aggregation and thrombus formation at the site of an atherosclerotic plaque rupture. The rupture of an atherosclerotic plaque or acute erosion of the endothelial layer can result in the formation of a pathological arterial thrombus, leading to the exposure of highly reactive subendothelial matrix proteins, such as collagen and von Willebrand factors (vWF) [1].

Arterial thrombosis is responsible for myocardial infraction and ischemic stroke, two of the leading causes of death globally. They are often treated pharmacologically with antiplatelet drugs, due to the fundamental role of blood platelets in the development of atherothrombosis [2].

Various in vitro and in vivo analyses indicate that certain dietary components with antiplatelet and anticoagulant properties, such as phenolic compounds, may also play a valuable role in the prophylaxis and therapy of cardiovascular diseases (CVDs) [3,4,5,6]. For example, isorhamnetin and two of its derivatives isolated from the phenolic fraction of sea buckthorn fruit, viz. 3-*O*-beta-glucoside-7-*O*-alpha-rhamnoside and 3-*O*-beta-glucoside-7-*O*-alpha-(3″′-isovaleryl)-rhamnoside, have been found to have antioxidant and anti-aggregation potential against washed blood platelets, and anticoagulant potential against plasma [4]; however, their mechanisms of action in whole blood remain poorly understood. Therefore, the aim of the present study was to determine the biological activity of isorhamnetin (compound **1**), 3-*O*-beta-glucoside-7-*O*-alpha-rhamnoside (compound **2**) and 3-*O*-beta-glucoside-7-*O*-alpha-(3″′-isovaleryl)-rhamnoside (compound **3**), (Figure 1) using human washed blood platelets and human whole blood in vitro.

Isorhamnetin is a flavonol, representing one of the most important active ingredients in the fruit of Hippophae rhamnoides L., which has a broad pharmacological effect. Isorhamnetin and its derivatives have an effect on the protection of the circulatory system and cerebral vessels, with anti-cancer, anti-inflammatory, antioxidant, organ, anti-obesity effects, etc.

The anti-platelet and anticoagulant potential was evaluated using the following three key analytical approaches: (A) flow cytometry, by measuring the cell-surface exposition of P-selectin (CD62P) and the existence of the active form of GPIIb/IIIa (PAC-1 binding), which was performed in resting or agonist (ADP or collagen)-stimulated blood platelets after incubation with tested compounds and plant fraction; (B) the thrombus-formation analysis system (T-TAS), which was used to determine the influence of the tested compounds and plant fraction on thrombus formation in whole blood; (C) colorimetry, which was used to assess the changes of platelet adhesion to collagen after incubation with the tested compounds and plant fraction. The activity of the tested compounds and phenolic fraction was compared to that of a commercial product, Aronox (*Aronia melanocarpa* berry extract with anti-platelet properties) [8,9].

## 2. Results

The results of the T-TAS test indicate that the AUC_10_ of the tested phenolic compounds (compound **1**, **2** and **3**; 50 µg/mL) was markedly reduced compared to control values (Figure 2 and Figure 3).

Figure 3, Figure 4, Figure 5 and Figure 6 present the markers of blood platelet activation measures with flow cytometry. While changes in blood platelet activation were noted in whole blood treated with all the tested phenolic compounds (**1**–**3**) and the phenolic fraction from sea buckthorn fruits, these changes were not always statistically significant (Figure 3, Figure 4, Figure 5 and Figure 6). The significant inhibition of the exposition of P-selectin was observed only in the highest concentration (50 µg/mL) in compounds **2** and **3** in 10 µM ADP-activated blood platelets. In addition, at a concentration of 50 µg/mL, compound **2** significantly reduced the exposure of GPIIb/IIIa on blood platelets activated by 20 µM ADP or by collagen (Figure 3, Figure 4, Figure 5 and Figure 6).

Additionally, the anti-adhesive activity of the phenolic fraction from sea buckthorn fruits and phenolic compounds (**1**–**3**) was studied. The non-activated blood platelets and thrombin-activated platelets demonstrated significantly lower collagen adhesion following incubation with compounds **1**, **2**, and **3** (5 and 50 µg/mL), as well as with the phenolic sea buckthorn fruit fraction (Figure 7).

Table 1 compares the effects of isorhamnetin and its derivatives (50 µg/mL) on hemostasis in whole blood and washed blood platelets.

The strongest anti-platelet and anti-coagulant potential was demonstrated by compound **2**, which inhibited the exposition of GPIIb/IIIa and P-selectin on blood platelets in whole blood. It also demonstrated anti-adhesion properties in washed blood platelets. Compound **2** demonstrated the strongest anti-adhesion effect of both non-activated and thrombin-activated platelets. Compound two also had anti-coagulant potential in whole blood, as measured by T-TAS. The lowest biological activity was demonstrated by compound **1**, which demonstrated only anti-adhesive activity and anticoagulant activity, but did not have any effect on biomarkers of platelets activation measured by flow cytometry. The aronia berry extract (50 µg/mL, positive control) also showed anti-platelet activity, for example decreasing PAC-1 binding in blood platelets activated by 20 µM ADP and collagen; it also reduced the exposure of GPIIb/IIIa on blood platelets activated by 10 µM ADP, 20 µM ADP and collagen (data not presented).

## 3. Discussion

Four main classes of antiplatelet drugs are currently used in the therapy of cardiovascular diseases. The first type are cyclooxygenases 1 (COX1) inhibitors such as acetylsalicylic acid, which blocks thromboxane A_2_ production [2]. Other drugs are used to inhibit the P_2_Y_12_ receptors for ADP; these are divided into the following two groups: thienopyridines, such as clopidogrel, and nucleoside-nucleotide derivates, such as cangrelor. The third group comprises proteinase-activated receptor 1 (PAR1) antagonists such as vorapaxar, which was developed to target thrombin; the platelet reaction to thrombin is mainly mediated by the following two G protein-coupled PARs: PAR1 and PAR4. Finally, GPIIb/IIIa inhibitors can be used, such as abciximab, a human antigen-binding fragment of mouse monoclonal antibody, tirofiban, a nonpeptidic small molecule mimicking the fibrinogen binding site, and eptifibatide, a cyclic heptapeptide with lysine-glycine-aspartic acid and a motif mimicking the fibrinogen binding sequence within GPIIb/IIIa. Unfortunately, in numerous cases, the use of these drugs is connected with an elevated risk of bleeding [2].

It is known that phenolic compounds demonstrate a range of pro-health effects on the human body, including various anti-inflammatory, antilipidemic, antiviral and antibacterial, hepatoprotective and cardioprotective properties [10]. These properties allow them to be used as active ingredients in various nutraceutical, cosmetic and even pharmaceutical preparations, and can be found in red wine, tea, coffee and medical herbs, as well as in fruits and vegetables [10]. In addition, phenolic compounds, including flavonoids isolated from various parts of plants possess anti-platelet activity. For example, they may inhibit P-selectin exposition, which has a crucial role in the connection of inflammation and hemostasis. P-selectin is an integrin belonging to heterodimeric transmembrane receptors, formed by noncovalent association of *α* and *β* chains; its main role is to meditate interactions with adhesion molecules on other cells and extracellular matrix molecules [1].

Flavonols are a class of flavonoids that have the 3-hydroxyflavone backbone. Their diversity stems from the different positions of –OH groups. Isorhamnetin is a flavonol isolated from the leaves, flowers and fruits of E. rhamnoides, Ginkgo biloba and other plants. It shows a broad range of pharmacological activity against cardiovascular disease, tumors and some neurovegetative diseases, such as Alzheimer’s disease. Its therapeutic effect against cardiovascular and cerebrovascular diseases is based on its ability to protect endothelial cells, and its anti-atherosclerosis, anti-hypotension and anti-thrombosis activity [10,11]. It can also improve nerve function and enhance memory. It has also demonstrated a bacteriostatic effect and has been proposed as a candidate for antibacterial drug research [11]. Isorhamnetin has also demonstrated some anti-tumor effects, manifested in the inhibition of human cervical cancer cell, lung cancer cells, breast cancer cell and liver cancer cells. It inhibits the proliferation of tumor cells by inducing apoptosis and the regulation of tumor suppressor genes and signal pathways [10,11].

The anti-inflammatory effects of isorhamnetin derive from its ability to regulate the production of inflammatory mediators such as cytokines, and to inhibit the NF-ĸB (nuclear factor kappa-light-chain-enhancer of activated B cells) pathway, which regulates a range of inflammatory molecules; indeed, its influence on NF-κB signaling is known to protect endothelial cells against inflammation and oxidative damage [11]. Isorhamnetin has also been found to reduce the risk of thrombosis by inhibiting collagen-stimulated platelet aggregation and various aspects of signal transduction; in addition, similarly to all flavonoids, it has been found to demonstrate strong antioxidant activity by scavenging radicals and inhibiting lipid peroxidation [10]. Moreover, isorhamnetin can inhibit cardiac hypertrophy and fibrosis by blocking the activation of the phosphatidylinositol 3-kinase-AKT (protein kinase B) signaling pathway. In in vitro studies, isorhamnetin attenuated cardiomyocyte hypertrophy in neonatal rat cardiomyocytes induced by angiotensin II [11].

Flavonoids also show some cardiovascular activity; this is mostly connected with their antiplatelet activity, i.e., preventing primary clot formation and inhibiting platelet aggregation. A structure-activity analysis attributed this antiaggregatory activity to the C-ring structure of flavonoids, as non-methylated flavonoids with a double bond between C2 and C3 have the strongest antiaggregatory activity [10]. In addition, the antiplatelet activity of plants is often related to their flavonoid and phenolic content. The leaves from Melissa officinalis, whose main compounds are flavonoids, reduced ADP-induced platelet aggregation by up to 18% [12,13]. More details about biological properties, including pharmacological activity have been described by Gong et al. [11].

Sea buckthorn fruits have been found to be a safe and valuable source of phenolic compounds such as flavonoids. Isorhamnetin and two isorhamnetin derivatives, 3-O-beta-glucoside-7-O-alpha-rhamnoside (compound **2**) and 3-O-beta-glucoside-7-O-alpha-(3″′-isovaleryl)-rhamnoside (compound **3**), have been reported to have antioxidant, anti-platelet and anticoagulant properties against plasma and washed platelets in vitro [4]. The present study examined the effects of these three flavonoids (compounds **1**–**3**) and a phenolic fraction from sea buckthorn fruits on selected aspects on hemostasis in whole blood. It is the first study to compare the effects of these three phenolic compounds on inhibiting thrombus formation and their anticoagulant properties in whole blood measured by T-TAS; this approach imitated in vivo conditions to assess whole blood thrombogenicity.

A significant role in clotting is played by GPIIb/IIIa signaling, with inside-out GPIIb/IIIa signaling being activated during platelet activation. However, ligand-bound GPIIb/IIIa can result in outside-in signaling events that mediate cytoskeletal reorganization. The initial arterial thrombus is reinforced by thrombin generation, which further increases platelet activation and activates coagulation, leading to greater stabilization of the fibrin mesh [2]. In the present study, compound **2** (50 µg/mL) was found to reduce GPIIb/IIIa exposition in platelets activated by 20 µM ADP. The GPIIb/IIIa complex is recognized by PAC-1, which promotes platelet aggregation. These findings suggest that the compound has an inhibitory effect on platelet aggregation. In addition, all the tested flavonoids (i.e., compounds **1**–**3**) also inhibited the activation of platelets stimulated by collagen.

Compounds **2** and **3** also appeared to inhibit P-selectin exposition in stimulated platelets, again indicating anti-platelet activity. The tested flavonoids also inhibited the adhesion of washed platelets to collagen, as indicted colorimetrically. Compound **3** (50 µg/mL) showed the greatest anti-adhesive properties against thrombin-stimulated platelets. In addition, our earlier results reported that compound **3** has inhibitory action against thrombin-stimulated human blood platelet aggregation (using washed platelets). It may suggest that this compound could modulate platelet activation by interfering with thrombin receptors on platelets [4].

Previous studies have also found isorhamnetin to have strong antiplatelet potential. For example, isorhamnetin administered at 1–100 µM significantly inhibited platelet aggregation induced by collagen and thrombin receptor activator peptide [14]. It also reduced the level of ATP (adenosine triphosphate) in collagen-stimulated platelets [14]. Stainer et al. [15] also demonstrated the antithrombotic activity of isorhamnetin in in vitro studies. The analysis included biomarkers of platelets activation, such as P-selectin exposure, fibrinogen binding and aggregometry. Isorhamnetin inhibited the aggregation induced by collagen and binding platelets to fibrinogen. The obtained results confirmed the antithrombotic and anticoagulant activity of isorhamnetin [15]. Yun et al. [16] analyzed the anti-atherosclerosis effect of isorhamnetin isolated from *Hippophae rhamnoides* L. in an in vivo study. Six-week old male C57BL/6J mice were dived in four groups. Isorhamnetin in a dose of 20 mg/kg of body weight was orally administrated. The study included the analyses of atherosclerotic lesion macrophage accumulation and apoptosis and lipid levels in serum. The results demonstrated isorhamnetin’s ability to decrease the occurrence of macrophage-induced apoptosis in atherosclerotic lesions, which confirmed the protective effect of isorhamnetin on the cardiovascular system [16].

Compound **2** was found to demonstrate stronger anti-platelet and anti-coagulant potential than compound **3** and isorhamnetin in whole blood; these differences may be due to their chemical structure. Our earlier results also reported that compound **2** has stronger antioxidant activity than compound **3** and isorhamnetin [4]. The most significant innovative finding of our experiment is that the tested flavonoids often demonstrated similar, or stronger, anti-platelet and anti-coagulant properties than the whole phenolic fraction. In addition, it is important to note that the flavonoids and phenolic fraction were administered in two concentrations, 5 and 50 µg/mL, the lower of which may correspond to the physiological concentration of plant-derived phenolic compounds available after oral supplementation.

Although the study demonstrated anticoagulant and antiadhesive activity of the isorhamnetin derivate, there were some weak points. The obtained results demonstrated the antiadhesive activity in all the tested compounds; however, there was limited confirmation in the measurement of biomarkers of platelets’ activation. In addition, only two concentrations were tested, which does not reveal the full range of potential effects. The antiplatelet effect was observed only in a higher concentration, which limited the use of isorhamnetin in potential therapy with oral administration. The results presented in this study are promising, but more detailed research should be conducted.

### Conclusions

An important and novel finding of the present study is that the flavonoids isolated from sea buckthorn fruits possess anticoagulant and anti-platelet (including anti-adhesive) potential against both washed blood platelets and whole blood. Of the tested compounds, compound **2** exhibited stronger properties than isorhamnetin, probably due to the presence of the glucose and rhamnose moieties, which may be involved in its interactions with platelets and the coagulation system. Similarly, the lower activity of compound **3** may be explained by the presence of the isovaleryl group, which increases the hydrophobicity of the compound, and may sterically hinder such interactions. Different actions of the tested flavonoids on platelet activation may also depend on their binding ability to various receptors on blood platelets. However, the molecular mechanism of their anti-platelet potential required further additional studies, including in vitro and in vivo experiments. In addition, their true effects on CVDs should be verified by conducting in vivo studies.

## 4. Materials and Methods

### 4.1. Chemicals

Flow cytometry reagents included CD61 PerCP (RUU-PL7F12, cat. No 3473408), PAC-1 FITC (Mouse BALB/c IgM, κ cat. No 34507), CD62-PE (P-selectin, Mouse BALB/c IgG1 κ, cat. No 550888), CellFIX (10 concentrated, cat no 340181), which were acquired from Becton Dickinson (Franklin Lakes, NJ, USA). The PL-chip (cat. No 18002), reservoir kit for PL-chip (cat. No 18003) and BAPA tubes (3 mL) and other equipment needed for the T-TAS were purchased from Bionicum (Warsaw, Poland). Collagen and ADP were obtained from Chrono-Log Corporation (Havertown, PA, USA). Dimethylsulfoxide (DMSO), isorhamnetin, Triton X-100 and p-nitrophenylphosphate were purchased from Sigma-Aldrich (Saint Louis, MO, USA). A stock solution of Aronox (Aronia melanocarpa berry extract, Agropharm Ltd., Warsaw, Poland) was prepared in water. The remaining reagents, including NaCl, Tris, NaOH and glucose were obtained from POCh (Gliwice, Poland).

### 4.2. Plant Material

Sea buckthorn fruits (*E. rhamnoides* (L.) A. Nelson) were acquired from a horticultural farm in Sokółka (Podlaskie Voivodeship, Sokółka, Poland (53°24′ N, 23°30′ E). The frozen fruits were ground, freeze-dried (Gamma 2-16 LSC, Christ, Osterode am Harz, Germany) and stored in a refrigerator. All plant studies involved in the research were carried out in accordance with relevant institutional, national or international guidelines. The entire section was previously described in Olas et al. [5] and Skalski et al. [4].

### 4.3. Preparation and Quantification of the Phenolic Fraction from Sea Buckthorn Fruits

Freeze-dried sea buckthorn fruits were subject to cold extraction with 80% methanol, then assisted with ultrasonic treatment and further extracted by boiling 80% methanol under pressure to remove the organic solvent. It was further purified by SPE on a short C18 column. The tested phenolic fraction from sea buckthorn fruits was analyzed using the Thermo Ultimate 300RS (Thermo Fischer Scientific, Waltham, MA, USA) chromatographic system. Methods were previously described by Olas et al. [5]. Its main components are isorhamnetin glycosides, acylated isorhamnetin glycosides, triterpenoids and acylated triterpenoids. More details about the chemical content of the tested phenolic fractions can be found in [5].

### 4.4. Isolation and Structure Determination of Flavonoids

The use of the phenol fraction purification procedure allowed for isolation and identification based on the UHPLC-MS analysis and the comparison of molecular weights and retention times with those described in the literature, compound **1** isorhamnetin and compound **2**: 3-O-beta-glucoside-7-O-alpha-rhamnoside.

On the other hand, compound **3**: 3-O-beta-glucoside-7-O-alpha-(3″′-isovaleryl) -rhamnoside isolated from the phenol fraction, as stated by Żuchowski et al. [7] was an unknown compound, not described in the literature, and its structure was determined by spectral methods, including NMR.

### 4.5. Stock Solutions of Tested Compounds and Plant Fraction

Stock solutions of sea buckthorn phenolic fraction, and of isorhamnetin (compound **1**) and its two derivatives were made in 50% DMSO; the derivatives were isorhamnetin 3-O-beta-glucoside-7-O-alpha-rhamnoside (compound **2**) and isorhamnetin 3-O-beta-glucoside-7-O-alpha-(3″′-isovaleryl)-rhamnoside (compound **3**) [6]. The DMSO concentration in the final samples did not exceed 0.05% and its effects were checked in every experiment.

### 4.6. The Samples of Blood

Fresh human blood was collected in the L. Rydygier hospital in Lodz, Poland. All donors were healthy volunteers, none were smokers or reported taking drugs. Blood was collected in tubes with CPDA anticoagulant (citrate/phosphate/dextrose/adenine; 8.5:1; *v*/*v*; blood/CPDA).

The blood used in the flow cytometry and T-TAS assays was incubated (30 min, at 37 °C) with *E. rhamnoides* (L.) fraction, isorhamnetin and its derivatives at final concentrations of 5 and 50 μg/mL.

#### Confirmation by Human Participants

All experiments were approved by the University of Lodz Committee for Research on Human Subjects and carried out under permission number 3/KBBN-UŁ/II/2016. 

We confirm that all experiments were performed in accordance with relevant guidelines and regulations. All donors were informed about the purpose of the study and gave their informed consent to participate.

### 4.7. Isolation of Blood Platelets

Fresh human blood (200× *g*, 12 min, at 25 °C) was centrifuged. Platelet-rich plasma (supernatant) was collected in Falcon tubes and centrifuged at 350× *g* for 15 min at 25 °C. The obtained platelet pellets were washed and re-suspended in Barber’s buffer (0.14 M NaCl, 0.014 M Tris, 5 mM glucose, pH 7.4) [17]. Further details of the method are provided by Wachowicz and Kustron [17]. The platelet concentrations in the suspensions used in the experiments ranged from 2 to 2.5 × 10^8^/mL, as indicated spectrophotometrically [18]. Blood platelet suspensions were incubated (30 min, at 37 °C) with *E. rhamnoides* (L.) fraction, isorhamnetin and its derivatives at final concentrations of 5 and 50 μg/mL.

### 4.8. Markers of Hemostasis

#### 4.8.1. Platelet Adhesion to Collagen

The test was performed to measure the activity of the platelet exoenzyme acid phosphatase. The platelets were first dissolved in Triton X-100. The phosphatase substrate p-nitrophenylphosphate was then added. The resulting formation of p-nitrophenol was determined spectrophotometrically at a wavelength λ = 405 nm. Finally, a color was obtained by adding 2M NaOH. The absorbance of the control (which included only blood platelets with Barber’s buffer) was expressed as 100%. The method is fully described in Bellavita et al. [19].

#### 4.8.2. Flow Cytometry Analysis

Changes in platelet activation were determined using an LSR II flow cytometer (Becton Dickinson, San Diego, CA, USA). Whole blood was incubated with the test compounds or phenolic fraction and platelet activators (ADP and collagen). The samples were then diluted with PBS with Mg^2+^ ions. Antibodies (CD61/PerCP; CD62/PE and PAC-1/FITC) were added to the cytometry tubes. The platelets were fixed in CellFix and incubated for one hour at 37 °C. The blood platelets were distinguished from other blood cells based on a forward light scatter (FCS) vs. side light scatter (SSC) plot on a log/log scale (first gate) and by positive staining with monoclonal anti-CD61/PerCP antibodies (second gate). In each sample, the percentages of CD62P-positive and PAC-1-positive platelets were measured. FlowJo software (Becton Dickinson, San Diego, CA, USA) was used to analyze the obtained results. The precise details of the method are described by Rywaniak et al. [20].

#### 4.8.3. Total Thrombus-Formation Analysis System (T-TAS)

The plate plug formation process was determined using a real-time hydrodynamic model. Whole blood (400 µL) was incubated with the tested fraction, isorhamnetin or its two derivatives (30 min, 37 °C). Next, 350 µL of blood was drawn for analysis. The results were obtained as AUC_10_ (Area Under the Curve) using a PL chip. A detailed description of the method can be found in Hosokawa et al. [21].

### 4.9. Data Analysis

The Q-Dixon test was used to discard uncertain data. All values are presented as means ± SD. N—number of donors. Statistical analysis was carry out using Statistica 13.1. Normal distribution of data was verified through normal probability plots and homogeneity of variance was confirmed by Levene’s test. Differences within and between groups (depending on Levene’s test) were studied using one-way ANOVA or Kruskal–Wallis test. As a post hoc to one-way ANOVA, we used Tuckey’s test.

## Figures and Tables

**Figure 1 molecules-27-04429-f001:**
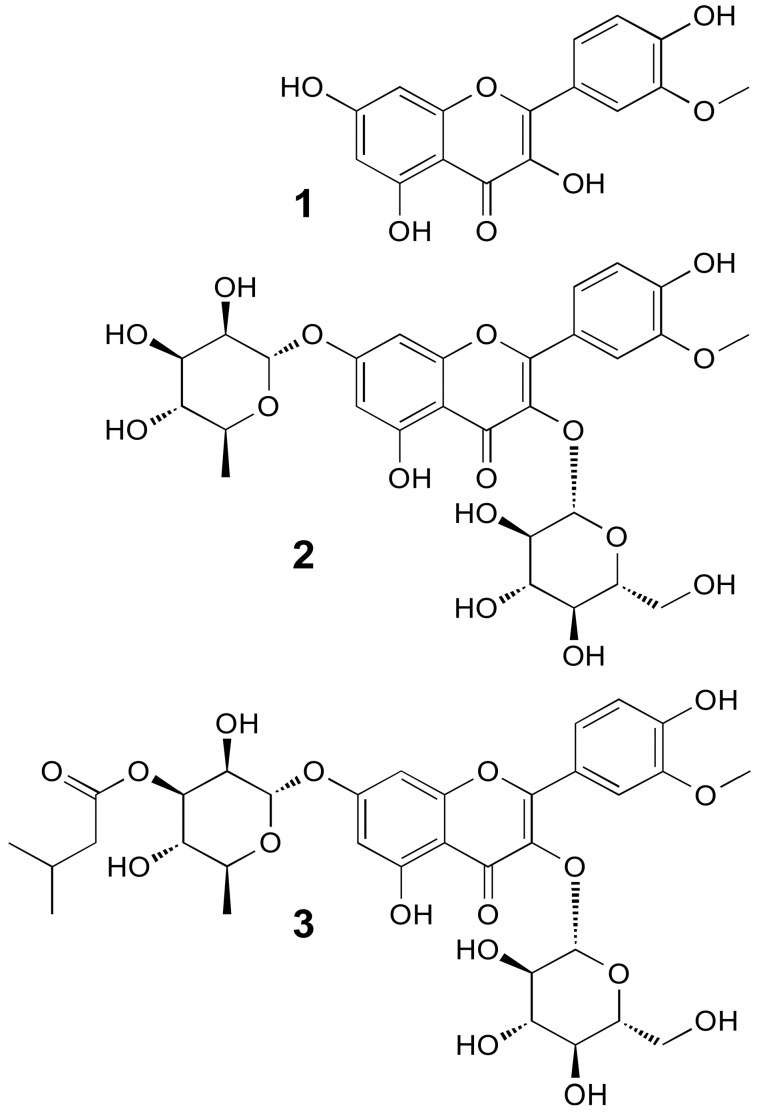
Chemical structure of isorhamnetin (compound **1**) and its derivatives: compound **2** (3-*O*-beta-glucoside-7-*O*-alpha-rhamnoside) and compound **3** (3-*O*-beta-glucoside-7-*O*-alpha-(3″′-isovaleryl)-rhamnoside) isolated from the phenolic fraction of *E. rhamnoides* (L.) A. Nelson fruits [4,7].

**Figure 2 molecules-27-04429-f002:**
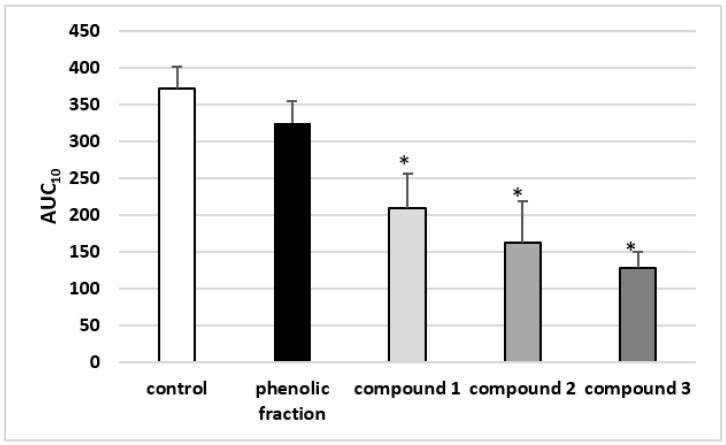
Effects of isorhamnetin, its derivatives and the phenolic fraction of *E. rhamnoides* (L.) A. Nelson fruits (50 µg/mL; 30 min) on the T-TAS using the PL-chip in whole blood samples. Whole blood samples were analyzed by the T-TAS at the shear rates of 1000 s^−1^ on the PL-chips. The area under the curve (AUC_10_) in PL are shown. Data represent the means ± SD of six healthy volunteers. * *p* < 0.05 vs. control.

**Figure 3 molecules-27-04429-f003:**
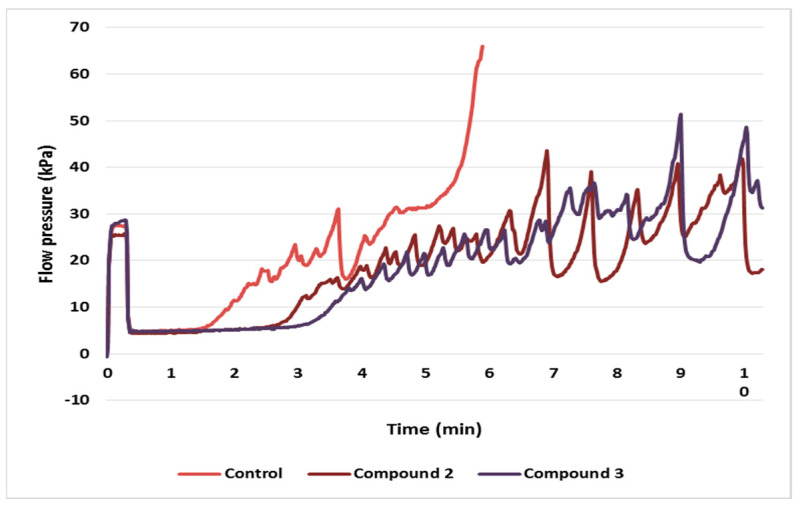
Flow pressure analysis reflect the platelet thrombus formation process using the PL-chip in whole blood (control-blood treated without derivatives of isorhamnetin; blood treated with 50 µg/mL derivatives of isorhamnetin: compound **2** and **3** within 10 min.

**Figure 4 molecules-27-04429-f004:**
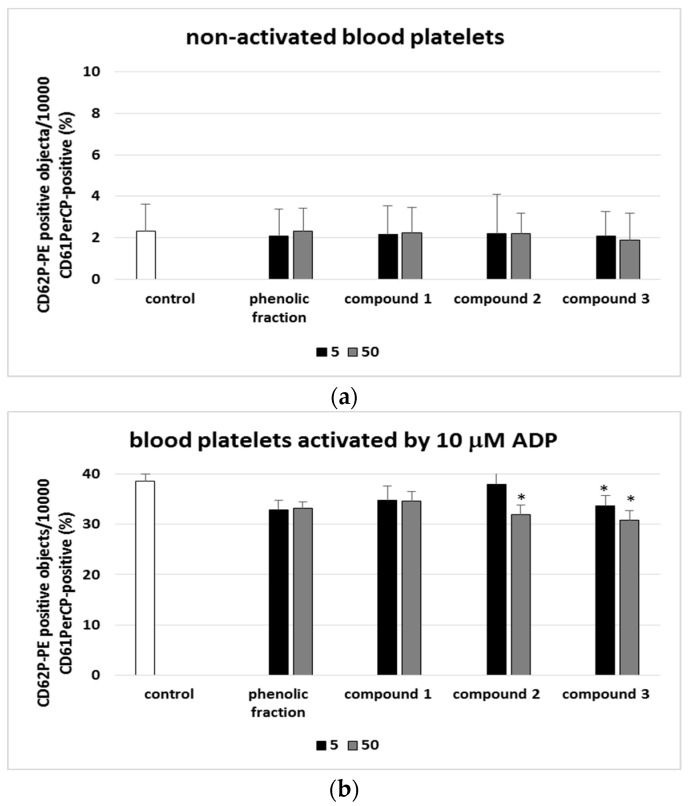
Effects of isorhamnetin, its derivatives and the phenolic fraction of *E. rhamnoides* (L.) A. Nelson fruits (5 and 50 µg/mL; 30 min) on the exposure of P-selectin on resting (**a**) or agonist-stimulated blood platelets: 10 µM ADP (**b**), 20 µM ADP (**c**) and 10 µg/mL collagen (**d**) in whole blood samples. The blood platelets were distinguished based on the exposition of CD61/PerCP. For each sample, 10,000 CD61-positive objects (blood platelets) were acquired. For the assessment of P-selectin exposition, samples were labeled with fluorescently conjugated monoclonal antibody CD62P. Results are shown as the percentage of platelets expressing CD62P. Data represent the means ± SD of 5 healthy volunteers. * *p* < 0.05 vs. control.

**Figure 5 molecules-27-04429-f005:**
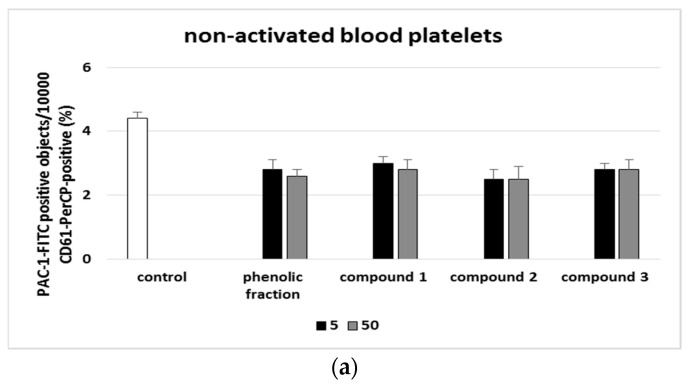
Effects of isorhamnetin, its derivatives and the phenolic fraction of *E. rhamnoides* (L.) A. Nelson fruits (5 and 50 µg/mL; 30 min) on the exposure of the active form of GPIIb/IIIa on resting (**a**) or agonist-stimulated blood platelets: 10 µM ADP (**b**), 20 µM ADP (**c**) and 10 µg/mL collagen (**d**) in whole blood samples. The blood platelets were distinguished based on the exposition of CD61. For each sample, 10,000 CD61-positive objects (blood platelets) were acquired. For the assessment of GPIIb/IIIa exposition, samples were labeled with fluorescently conjugated monoclonal antibody PAC-1/FITC. Results are shown as the percentage of platelets binding PAC-1/FITC. Data represent the means ± SD of 5 healthy volunteers. * *p* < 0.05 vs. control.

**Figure 6 molecules-27-04429-f006:**
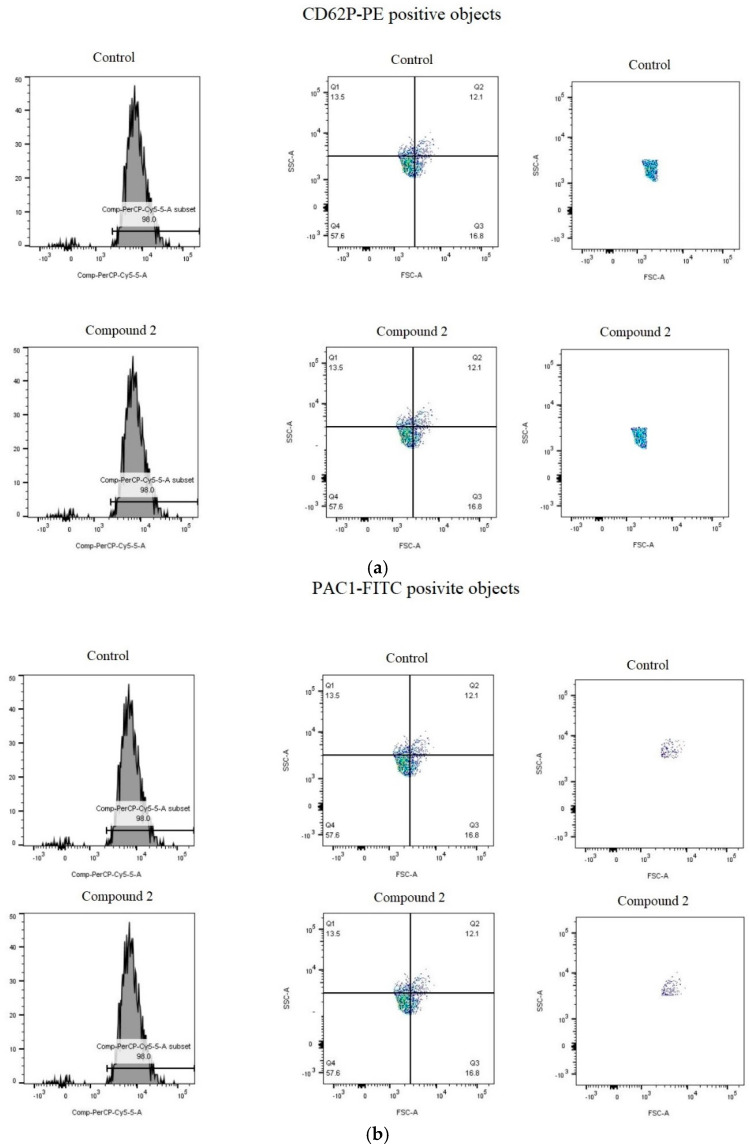
Effects of compound **2** (50 µg/mL; 30 min) on the exposure of P-selectin (**a**) and the active form of GPIIb/IIIa (**b**) in platelets stimulated by 10 µg/mL collagen in whole blood samples. Figure demonstrates selected diagrams.

**Figure 7 molecules-27-04429-f007:**
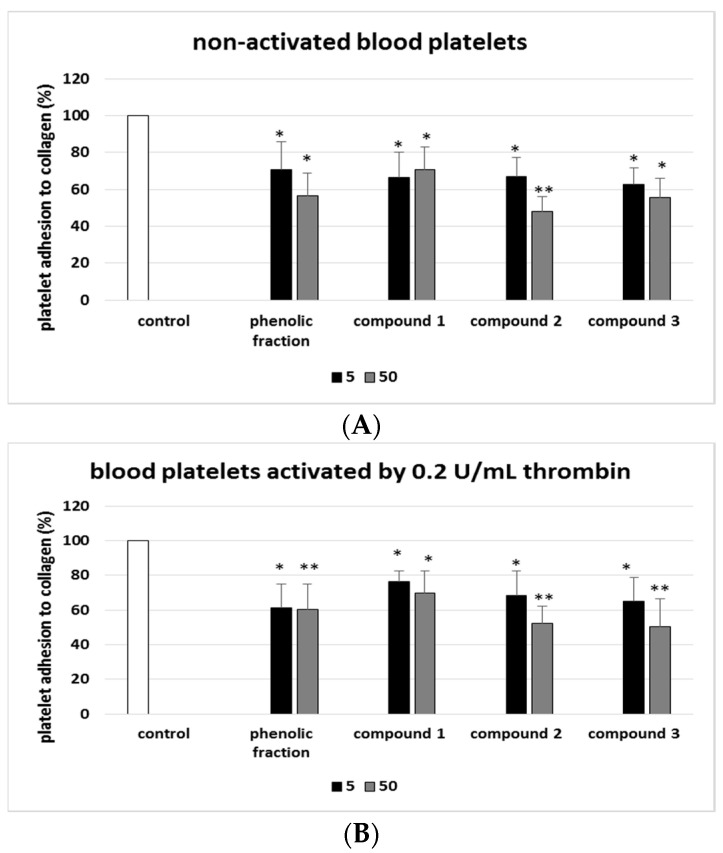
Effects of isorhamnetin, its derivatives and the phenolic fraction of *E. rhamnoides* (L.) A. Nelson fruits (5 and 50 µg/mL; 30 min) on the collagen adhesion of resting (**A**) and thrombin-activated platelets (**B**). Blood platelets not treated with phenolic compounds/plant fraction were used as control samples (positive control). Adhesion is expressed as a percentage of the control samples (100%). Data represent means ± SD of six healthy volunteers (experiments performed in triplicate). Action of phenolic compounds and phenolic fraction was compared to control: * *p* < 0.05, ** *p* < 0.01.

**Table 1 molecules-27-04429-t001:** Comparative effects of isorhamnetin and its derivatives (at 50 µg/mL) on parameters of hemostasis compared to control (whole blood or washed blood platelets without treated with phenolic compounds).

Tested Phenolic Compound	Anti-Adhesive Activity		Anticoagualnd Activity	Antiplatelet Activity (Measured by Exposition of GPIIb/IIIa and P Selectin)		
	Inhibition of Adhesion of Non-Activated Platelets to Collagen (%)	Inhibition of Adhesion of Thrombin-Activated Platelets to Collagen (%)	Inhibition of Thrombus Formation (%)	Inhibition of Exposition of GPIIb/IIIa on Platelets Activated by 20 µM ADP (%)	Inhibition of Expositionn of GPIIb/IIIa on Platelets Activated by Collagen (%)	Inhibition of Exposition of P Selectin on Platelets Activated by 10 µM ADP (%)
Compound **1**	18.4 ± 10.4	26.7 ± 7.8	42.4 ± 10.4	No effect	26.7 ± 4.5	No effect *v*
Compound **2**	54.5 ± 11.7	49.4 ± 10.1	61.4 ± 12.3	28.4 ± 8.2	32.4 ± 5.1	21.4 ± 3.8
Compound **3**	48.2 ± 12.5	50.9 ± 12.2	65.2 ± 11.0	No effect	26.1 ± 4.2	25.5 ± 4.1

## Data Availability

Not applicable.

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
