# Peer review of "Antiplatelet and Anticoagulant Activity of Isorhamnetin and Its Derivatives Isolated from Sea Buckthorn Berries, Measured in Whole Blood"

_molecules, 2022, doi:10.3390/molecules27144429_

Round 1
Reviewer 1 Report
The article ”Antiplatelet and anticoagulant activity of isorhamnetin and its derivatives isolated from sea buckthorn berries, measured in whole blood” present an interesting subject. The research seem to be performed correctly, but there are some aspects that need corrections.
Introduction
The authors say that the phenolic compounds have been studied in various analyses in vitro and in vivo, but they present only a reference. My suggestion is to add more than one in order to sustain this point of view.
Results
The results section is really small. It is made almost entirely of figures and one table. There is no proper interpretation of the data. The authors should better develop this section. The figures are very big as to add volume to the manuscript. The figure 1 quality is very poor. The authors should use a proper software to draw the chemicals presented.
The authors did not used any positive drug control in order to say that the compounds’ results deserve to be considered. The authors should provide the results obtained with a control drug like acetylsalicylic acid or clopidogrel.
- Make subsections for each determination
- T-TAS: there is an inconsistency between the figure 2 ant its legend: in the legend there are mentioned 2 concentration and, in the figure, only one. I understand from the text, that only the concentration of 50 μg/mL had some significant results, so correct the legend. Or, as for the next analyses there are presented the results for the 5 μg/mL concentration even there is nor a significant change, you could add the results for 5 μg/mL concentration here as well.
- If the compound 1 has also a significant result, why it is not in the figure 3?
- If it is possible, please make figure 6 more clear.
- table 1: I suggest to reorganized it three sections. antiadhesive action, anticoagulant action, antiplatelet action each with their subsections and mention in the legend that the results are vs control. In this way you can keep only the numbers or the mention NE (no effect) for each compound, without the text. it will easier to read in this way.
Discussion
Although I appreciate the authors did not exaggerate with the references, I am sure that there are more studies to support the isorhamnetin effects. The discussion section is many aspects out of scopus. It should focus on integrating the original results with similar works in literature.
Page 11, Row 170 the authors should use nonproprietary names, like acetylsalicylic acid and not aspirin.
The authors say that the lower concentration correspond to the physiological concentration of plant-derived phenolic compounds available after oral supplementation, but that concentration has significant results only for antiadhesive action, so what will be the necessary dose for the anticoagulant and anti-platelet actions?
The discussion should point out the weak points of the research and that the effects appear at high concentrations. The authors should be more objective in their interpretation of the data.
The final paragraph of discussion is actually the conclusion, so you can name it.
Author Response
The article ”Antiplatelet and anticoagulant activity of isorhamnetin and its derivatives isolated from sea buckthorn berries, measured in whole blood” present an interesting subject. The research seem to be performed correctly, but there are some aspects that need corrections.
Introduction
The authors say that the phenolic compounds have been studied in various analyses in vitro and in vivo, but they present only a reference. My suggestion is to add more than one in order to sustain this point of view.
Response: We have added more references. Now, it is “3-7”.
Results
The results section is really small. It is made almost entirely of figures and one table. There is no proper interpretation of the data. The authors should better develop this section. The figures are very big as to add volume to the manuscript. The figure 1 quality is very poor. The authors should use a proper software to draw the chemicals presented.
Response: We have changed the figures. Now, they are smaller. We developed results section by addicting more interpretation of this dates.
The authors did not used any positive drug control in order to say that the compounds’ results deserve to be considered. The authors should provide the results obtained with a control drug like acetylsalicylic acid or clopidogrel.
Response: We have added information about positive control. The activity of the tested compounds and phenolic fraction was compared to that of a commercial product: Aronox (Aronia melanocarpa berry extract with anti-platelet properties) [Olas et al., 2008; Skalski et al., 2021].
- Make subsections for each determination
Response: We have not prepared subsections, because, in this journal there is not subsection in the chapter of results.
- T-TAS: there is an inconsistency between the figure 2 ant its legend: in the legend there are mentioned 2 concentration and, in the figure, only one. I understand from the text, that only the concentration of 50 μg/mL had some significant results, so correct the legend. Or, as for the next analyses there are presented the results for the 5 μg/mL concentration even there is nor a significant change, you could add the results for 5 μg/mL concentration here as well.
Response: We have corrected the legend to figure 2. Now, it is only one concentration – 50 µg/mL. We have only analysed one concentration – 50 µg/mL, because T-TAS method is very expensive.
- If the compound 1 has also a significant result, why it is not in the figure 3?
Response: We have presented only flow pressure analysis reflect the platelet thrombus formation process using the PL-chip in whole blood for two new derivatives of isorhamnetin: compound 2 and 3, because when we add compound 1 the figure will be difficult to read.
- If it is possible, please make figure 6 more clear.
Response: Unfortunately we unable to make figure 6 more clear, due to lack of access to special software. We only got access for a few weeks during the analysis.
- table 1: I suggest to reorganized it three sections. antiadhesive action, anticoagulant action, antiplatelet action each with their subsections and mention in the legend that the results are vs control. In this way you can keep only the numbers or the mention NE (no effect) for each compound, without the text. it will easier to read in this way.
Response: We reorganized the table to make it easier to read.
Discussion
Although I appreciate the authors did not exaggerate with the references, I am sure that there are more studies to support the isorhamnetin effects. The discussion section is many aspects out of scopus. It should focus on integrating the original results with similar works in literature.
Response: We add more similar results from original work into discussion.
Page 11, Row 170 the authors should use nonproprietary names, like acetylsalicylic acid and not aspirin.
Response: We have corrected. Now, it is “acetylsalicylic acid”.
The authors say that the lower concentration correspond to the physiological concentration of plant-derived phenolic compounds available after oral supplementation, but that concentration has significant results only for antiadhesive action, so what will be the necessary dose for the anticoagulant and anti-platelet actions?
Response: Thank you for the comment, but without additional studies we unable to suggest the necessary concentration for significant results in anticoagulant and anti-platelets. However, concentration should be higher then the one used in studies, but still should be possible to obtain also in ex vivo studies by supplementation.
The discussion should point out the weak points of the research and that the effects appear at high concentrations. The authors should be more objective in their interpretation of the data.
Response: We add soma weak points and concern about obtained results.
The final paragraph of discussion is actually the conclusion, so you can name it.
Response: We have named the final paragraph – conclusion.

Reviewer 2 Report
This review discussed the subject ‘Antiplatelet and anticoagulant activity of isorhamnetin and its 2 derivatives isolated from sea buckthorn berries, measured in 3 whole blood’ look very interesting, yet it needs to be improved in a scientific manner about the subject matter. Grammatical and typographical errors especially punctuations should be carefully rechecked.
Abstract: Line 19-20- Compound name (number), eg: Isorhamnetin (1)
The term flavonoids changed to flavonol or quercetin derivatives.
Introduction:
Figure 1- use proper chemical draw software such as chem draw to draw the structure of isorhamnetin and its derivatives properly.
Should write a short introduction (one paragraph) on the chemistry and pharmacology of isorhamnetin and its derivatives.
Discussion:
Rather than using the term flavonoids, perhaps the authors can be more precise by using the proper term (flavonols or quercetin derivatives) to address the phytochemical.
Focus on the pharmacological aspect of this type of flavonoid. Detail more on the structure-activity relationship of isorhamnetin (1) and its derivatives (2-3) reported towards the activity exhibited with appropriate supporting references.
Conclusion?
The objectives of research achieved?
The future direction of this research?
Recommendation for improvement?
4. Materials and methods
4.4. Isolation and structure determination of flavonoids
Report the spectral data values (NMR and MS) obtained for the reported phytoconstituents (1-3), respectively.
Author Response
This review discussed the subject ‘Antiplatelet and anticoagulant activity of isorhamnetin and its 2 derivatives isolated from sea buckthorn berries, measured in 3 whole blood’ look very interesting, yet it needs to be improved in a scientific manner about the subject matter. Grammatical and typographical errors especially punctuations should be carefully rechecked.
Response: The language proofreading was done by a native speaker.
Abstract: Line 19-20- Compound name (number), eg: Isorhamnetin (1)
Response: We have corrected.
The term flavonoids changed to flavonol or quercetin derivatives.
Response: We have corrected.
Introduction:
Figure 1- use proper chemical draw software such as chem draw to draw the structure of isorhamnetin and its derivatives properly.
Response: Drawing changed to Tif format.
Should write a short introduction (one paragraph) on the chemistry and pharmacology of isorhamnetin and its derivatives.
Response: Short information about chemistry and pharmacology of isorhamnetin and its derivatives was added to the introduction.
Discussion:
Rather than using the term flavonoids, perhaps the authors can be more precise by using the proper term (flavonols or quercetin derivatives) to address the phytochemical.
Response: We have added more information about it.
Focus on the pharmacological aspect of this type of flavonoid. Detail more on the structure-activity relationship of isorhamnetin (1) and its derivatives (2-3) reported towards the activity exhibited with appropriate supporting references.
Response: We have added more information about it in the conclusion.
Conclusion?
Response: We have added the conclusion.
Round 2
Reviewer 1 Report
The authors made the required changes, therefore I agree with the publication of the article.
Author Response
We thank the Reviewer for the review of our manuscript.
Reviewer 2 Report
This revised manuscript still needs to be improved in a scientific manner about the subject matter. Again, I stressed that the grammatical and typographical errors especially punctuations should be carefully rechecked.
Abstract: Use either one term, flavonol or isorhamnetin derivatives to address the constituents.
Introduction: Line 73: Flavonol to O-methylated flavonol.
Line 74 Isorhamnetin (spelling)
Discussion: Line 217- is a flavonol
Line 244-249 The sentence should be scientifically profound. Focus the discussion on the flavonols rather than general aspects.
Should have a more profound scientific write-up on the outcome of pharmacological activity in relation to the chemical structure of isorhamnetin and its derivatives.
Line 312: Detail the aspects that can be further studied in terms of chemical and pharmacology.
3.1 Conclusion: Future direction of research.
4.4 Isolation and structure determination of flavonols.
Please provide the spectral data value for respective spectroscopy analyses of respective compounds reported.
Author Response
Dear Reviewer,
Response: I thank the Reviewer for helpful comments. Moreover, I agree with the comment of Reviewer, and this wrong statement was corrected. Unfortunately, I will not correct grammatical and typographical mistakes, because even a sworn translator of the English language, Ewa Lipowska-Szymańska from the Translation Agency PUŁAWY, cannot find them.
Abstract: Use either one term, flavonol or isorhamnetin derivatives to address the constituents.
Response: Form flavonol was used.
Introduction: Line 73: Flavonol to O-methylated flavonol.
Line 74 Isorhamnetin (spelling)
Response: Mistakes have been corrected.
Discussion: Line 217- is a flavonol
Response: The error has been corrected.
Line 244-249 The sentence should be scientifically profound. Focus the discussion on the flavonols rather than general aspects.
Should have a more profound scientific write-up on the outcome of pharmacological activity in relation to the chemical structure of isorhamnetin and its derivatives.
Response: We have added more information about it.
Line 312: Detail the aspects that can be further studied in terms of chemical and pharmacology.
Response: We have added more information about it.
3.1 Conclusion: Future direction of research.
Response: We have added more information about it.
4.4 Isolation and structure determination of flavonols.
Please provide the spectral data value for respective spectroscopy analyses of respective compounds reported.
Response: Spectral data for the respective spectroscopic analyzes of the respective compounds was provided in our earlier publication cited in the text.
- Żuchowski et al. Unusual isovalerylated flavonoids from the fruit of sea buckthorn (Elaeagnus rhamnoides) grown in Sokółka, Poland. Phytochem. 2019, 163, 178-186.